# Obesity-Associated Hepatic Steatosis, Somatotropic Axis Impairment, and Ferritin Levels Are Strong Predictors of COVID-19 Severity

**DOI:** 10.3390/v15020488

**Published:** 2023-02-09

**Authors:** Davide Masi, Elena Gangitano, Anna Criniti, Laura Ballesio, Antonella Anzuini, Luca Marino, Lucio Gnessi, Antonio Angeloni, Orietta Gandini, Carla Lubrano

**Affiliations:** 1Department of Experimental Medicine, Section of Medical Pathophysiology, Food Science and Endocrinology, Sapienza University of Rome, 00161 Rome, Italy; 2Department of Radiology, Anatomo–Pathology and Oncology, Sapienza University of Rome, 00185 Rome, Italy; 3Department of Mechanical and Aerospace Engineering, Sapienza University of Rome, 00185 Rome, Italy; 4Emergency Medicine Unit, Department of Emergency-Acceptance, Critical Areas and Trauma, Policlinico “Umberto I”, 00161 Rome, Italy; 5Department of Molecular Medicine, Sapienza University of Rome, 00161 Rome, Italy

**Keywords:** liver steatosis, liver density, obesity, SARS-CoV-2 infection, COVID-19, ferritin, growth hormone, IGF-1

## Abstract

The full spectrum of SARS-CoV-2-infected patients has not yet been defined. This study aimed to evaluate which parameters derived from CT, inflammatory, and hormonal markers could explain the clinical variability of COVID-19. We performed a retrospective study including SARS-CoV-2–infected patients hospitalized from March 2020 to May 2021 at the Umberto I Polyclinic of Rome. Patients were divided into four groups according to the degree of respiratory failure. Routine laboratory examinations, BMI, liver steatosis indices, liver CT attenuation, ferritin, and IGF-1 serum levels were assessed and correlated with severity. Analysis of variance between groups showed that patients with worse prognoses had higher BMI and ferritin levels, but lower liver density, albumin, GH, and IGF-1. ROC analysis confirmed the prognostic accuracy of IGF-1 in discriminating between patients who experienced death/severe respiratory failure and those who did not (AUC 0.688, CI: 0.587 to 0.789, *p* < 0.001). A multivariate analysis considering the degrees of severity of the disease as the dependent variable and ferritin, liver density, and the standard deviation score of IGF-1 as regressors showed that all three parameters were significant predictors. Ferritin, IGF-1, and liver steatosis account for the increased risk of poor prognosis in COVID-19 patients with obesity.

## 1. Introduction

Coronavirus disease (COVID-19) is an infectious disease that still poses a global pandemic challenge [1], and the prognostic factors responsible for its severity have yet to be thoroughly investigated [2,3,4]. Severe acute respiratory syndrome coronavirus 2 (SARS-CoV-2) primarily affects the respiratory tract, but it can also cause multi-organ dysfunction due to the widespread presence of the angiotensin-converting enzyme–2 (ACE–2) receptor, which is the entry site for the virus [5].

Possible effects of SARS-CoV-2 infections on the endocrine system, including changes in the function of the thyroid [6], pancreas, adrenal glands, and gonads have been increasingly reported [7,8]. However, only a few clinical studies have explored the potential link between SARS-CoV-2 infections and the growth hormone (GH)/insulin-like growth factor–1 (IGF-1) axis [9,10] and between liver steatosis and COVID-19 severity [11,12,13]. It is well known that GH deficiency states are associated with severe non-alcoholic steato-hepatitis (NASH) [14,15] and that obesity is generally associated with low GH levels and hepatic steatosis [16,17]. SARS-CoV-2 seems to reduce the insulin/IGF signaling in lung and metabolic tissues (e.g., liver, adipose tissue) [18]. Recently, it has also been reported that the farnesoid X receptor (FXR) may modulate ACE2 transcription in multiple human tissues [19]. Furthermore, FXR plays a pivotal role in regulating iron homeostasis [20] and interestingly transcriptomic profiles of liver biopsies revealed a strong relationship between IGF-1 expression and FXR signaling [21].

Interestingly, there is a similar pattern between ACE–2 receptor expression and age-related changes in growth hormone (GH) secretion [22]. Recently, speculations have been made about reduced GH action in patients with COVID-19 and obesity [23,24]. Individuals with obesity, particularly those who are older and have visceral adipose tissue (VAT) accumulation, are at higher risk of more severe COVID-19 complications [25], and reduced GH secretion is a hallmark of this population subgroup [16,26]. A preclinical study conducted in hamsters infected with SARS-CoV-2 confirmed that diet-induced obesity and NASH impair disease recovery [27]. In addition, a recent review further highlighted the close association of COVID-19 with hepatic metabolic dysfunction [28].

Obesity may also act as an effect modifier of smog-induced lung injury, and the concomitant presence of these two factors could better explain the higher virulence, faster spread, and more significant mortality in polluted areas [29]. Furthermore, GH has been identified as a candidate disease-modifying target in non-alcoholic fatty liver disease (NAFLD) because of its lipolytic and anti-inflammatory properties [17,30]. First, unequivocal evidence suggests that immune dysregulation is a core element in determining the severity of SARS-CoV-2 and the association with GH deficiency (GHD) in adulthood [23]. In addition, as GH is physiologically involved in developing and maintaining the immune system, its pharmacological replacement in GHD patients appears to positively influence their inflammatory status [31]. Furthermore, impairment of fibrinolysis associated with GHD may represent a further link between the impairment of the GH–IGF-1 axis and the severity of COVID-19, as it has been associated with both conditions [32,33]. Preventive measures based on an understanding of the pathophysiology of the disease may be of utmost importance to mitigate its spread. Several pieces of evidence supported a possible relationship between GHD and COVID-19 severity and have also shed light on the potential beneficial effects of treatment with recombinant GH on COVID-19 patients [31]. In this regard, recently published articles showed that among the hormonal changes in long COVID patients are reduced GH levels and possibly reduced insulin/IGF-1 signaling [34,35]. Considering all the above, a close relationship between ferritin metabolism, obesity, hepatic steatosis, and the GH/IGF-1 axis with COVID-19 severity is conceivable.

Therefore, the objective of the study was to evaluate if weight-related alterations of the GH/IGF-1 axis, liver attenuation on CT, and iron metabolism were present in COVID-19 patients upon their first admission to the emergency department and possibly related to COVID-19 severity, comparing individuals who progressed to a more severe form of the disease with individuals with a stable infection that did not require oxygen-supported intervention, intubation, or admission to intensive care units.

## 2. Materials and Methods

### 2.1. Study Design

This is an observational, single-center, retrospective study conducted according to the guidelines of the Declaration of Helsinki.

### 2.2. Study Population

A total of 143 COVID-19 patients admitted to the Emergency Medicine Department of the Polyclinic Umberto I in Rome between March 2020 and May 2021 were included in the study and provided verbal consent to participate. Serum samples were collected upon admission before starting any treatment and tested by the Laboratory Department. Inclusion criteria were as follows: (1) SARS-CoV-2 infection defined as the presence of at least two positive reverse transcriptase polymerase chain reaction results from nasopharyngeal swabs; (2) chest CT imaging suggestive of COVID-19 pneumonia; and (3) age >18 years. According to the WHO guidelines, patients were divided into four different groups according to the severity of pulmonary impairment in CT and respiratory failure [36]: patients with no CT alterations (Group 0–mild); patients with changes in CT scan requiring no oxygen support (Group 1–moderate); patients with CT scan plus oxygen supplementation (Group 2–severe) and patients with CT abnormalities plus intensive care unit (ICU) admission (Group 3–critical). The lung involvement, reported as the percentage of parenchyma affected by the disease, was established through the analysis of the chest CT by expert radiologists following a standardized procedure [37,38]. After the initial evaluation and management, patients were discharged in home isolation or were hospitalized in low, medium, or sub-intensive/intensive care units according to their medical needs. All patients were followed up to 60 days after emergency department admission. Patients were further grouped based on whether they experienced death/ARDS (acute respiratory distress syndrome) within two months of admission or not.

### 2.3. Measurements

Demographic characteristics, including race (which was self-reported), clinical history, and clinical presentation, were collected from the electronic medical records at admission and during hospitalization. Biochemical variables (such as white blood cell count, serum electrolytes, creatinine, ferritin, C-reactive protein (CRP), fibrinogen, D–dimer test, glucose, and liver parameters), arterial blood gas analysis with the corresponding PaO_2_/FiO_2_ ratio (P/F ratio), and the need for oxygen supplementation were also evaluated. Apart from the routine laboratory workup for COVID-19, we collected serum samples from patients with diverse presentations (from asymptomatic cases to the most severe forms), clinical trends, and outcomes. Samples were then transferred to the local laboratory and handled according to the local standards of practice. In doing so, we measured GH and IGF-1 levels within 3 h of admission to the emergency ward. Specifically, IGF-1 was assayed by an immunoradiometric assay after ethanol extraction (Diagnostic System Laboratories Inc., Webster, TX, USA). As serum IGF-1 levels highly depend on age and sex, we also normalized IGF-1 values for these two parameters. We expressed them as standard deviation scores (zSDSs), as previously described by Chanson et al. [39]. Hepatic steatosis index (HSI) was calculated as follows: *8 X(ALT/AST ratio)+BMI (+2, if female; +2, if diabetes mellitus)* [40] and Fibrosis–4 index for liver fibrosis (FIB-4) was calculated using the following formula: *age(years) X AST [U/L]/(platelets [10^9^/L] X (ALT [U/L])1/2)* [41]. A threshold value of <1.45 has a negative predictive value for the exclusion of extended fibrosis of 90%. A threshold value of >3.25 has a positive predictive value for the diagnosis of extensive fibrosis of 65%.

### 2.4. CT Imaging of the Liver

All examinations were performed using two multidetector CT scanners (Somatom Sensation 16 and Somatom Sensation 64; Siemens Healthineers, Erlangen, Germany). Scan parameters corresponded to the manufacturer’s recommended standard presets for a chest routine. Chest CT images were retrospectively analyzed for hepatic measurement at hospital admission. Two expert radiologists independently interpreted the images. First, each reader chose the CT scan that allowed the best visualization of the liver. Liver attenuation was defined as the mean attenuation of three regions of interest (ROIs) expressed in Hounsfield units (HUs). Two ROIs were placed in the anterior and posterior segments of right liver lobe and one ROI in the left liver lobe. Liver attenuation <48 HU was used as a cut-off for the diagnosis of hepatic steatosis [42].

### 2.5. Statistical Analysis

Statistical analysis has been performed using MedCalc^®^ Statistical Software version 20.111 (MedCalc Software Ltd., version 20.2, Ostend, Belgium; https://www.medcalc.org (accessed on 1 September 2022); and StatSoft, Inc. STATISTICA^®^ version 12, Helsinki, Finland; (Data analysis software system; www.statsoft.com (accessed on 1 September 2022). Distribution of continuous variables was tested with the Shapiro–Wilk test; linearity was established by visual inspection of a scatterplot. Data points greater or less than two standard deviations from the mean have been considered statistical outliers and excluded from all analyses.

Descriptive statistics (n, mean, SD) were calculated for continuous variables, whereas other variables were expressed as percentages and frequencies, as appropriate. Relationships between study variables were calculated using univariate regression analysis with Pearson or Spearman (Rho) coefficients for skewed data, with a two-tailed *p* < 0.05 indicating statistical significance.

Multivariate stepwise linear regression was used to evaluate the independent predictors of COVID-19 severity. The receiver operating characteristic (ROC) curve was assessed.

## 3. Results

A total of 143 patients were included in the study. Clinical parameters, biochemical tests, respiratory status, and hormonal evaluation recorded on admission are presented in Table 1. The mean age of participants was 60.63 ± 17.04 years and 75 patients (52.45%) were male. At the time of admission to the Emergency Department, 51 patients (35.66%) exhibited a P/F ratio <300, and 61 patients (42.66%) displayed severe lung involvement.

GH and IGF-1 levels were 0.92 ± 1.06 ng/mL and 97.2 ± 63.4 ng/mL, respectively. After normalization for age and sex, we found that the zSDS–IGF-1 was −2.62 ± 1.64.

### Predicors of COVID-19 Severity

The mean liver density was 50.7 ± 9.5 HU. Based on assessments of CT scans, hepatic steatosis was present in 49 patients (34%, HU_liver_: 40.04 ± 8.19) and was absent in 94 patients (66%, HU_liver_: 55.65 ± 4.57).

According to pulmonary involvement, patients were divided as follows: 17 (11.89%) in Group 0 without CT scan alterations or hypoxia; 26 (18.18%) in Group 1 with CT scan signs suggestive of pneumonia but without oxygen supplementation; 37 (25.87%) in Group 2 with CT scan signs suggestive of pneumonia and oxygen supplementation; and 63 (44.06%) in Group 3 with CT scan changes suggestive of ARDS requiring intensive care unit (ICU) admission [36].

One-way analysis of variance between groups, depicted in Table 2, showed that higher BMI is associated with more severe disease, confirming obesity as an adverse prognostic factor. In addition, the group of patients with worse prognosis had significantly higher levels of ferritin, fibrinogen, lactate dehydrogenase, and CRP but, at the same time, a lower hepatic attenuation, P/F ratio and significantly lower concentrations of GH, IGF-1, and zSDS–IGF-1. The main findings are presented in Figure 1.

A Student’s t-test for independent samples, shown in Table 3, was performed to assess the differences between patients with ARDS/death and all other patients, which constituted the control group. The test confirmed that higher values of indices estimating the amount of scarring and steatosis in the liver, as well as low levels of liver attenuation, IGF-1, and *zSDS–IGF-1* values, are associated with the need for *ventilation or occurrence of death in* COVID-19 patients.

A multivariate regression analysis to assess independent predictors of COVID-19 severity (groups 0–3) is presented in Table 4 and reveals that ferritin was positively correlated with the severity of the disease. In contrast, both hepatic attenuation and zSDS-IGF-1 were negatively correlated.

In addition, we built the receiver operating characteristic (ROC) curve with the Youden index to test the predictive value of serum IGF-1 as a continuous variable against 60-day outcomes (Figure 1). For this purpose, the optimal cut-off of serum IGF-1 < 64.91 ng/dl could discriminate patients with critical clinical conditions with a sensitivity and specificity of 64.1% and 69.7%, respectively (AUC 0.688, CI: 0.587 to 0.789, *p* < 0.001).

## 4. Discussion

Obesity is an established risk factor for severe COVID-19 outcomes, but the reasons for this association are not fully understood. Strands of research support the study of systemic inflammatory pathways in obesity-associated severe COVID-19 since significantly higher CRP, ferritin [44], and ESR values have been found [45]. Furthermore, central obesity, hypertension, and smoking habits seem to be associated with lower Ab titers following COVID-19 vaccination [46].

Moreover, available clinical data suggest a more aggressive course of SARS-CoV-2 infections in the elderly, males, and patients with obesity; therefore, in a previous article, we suggested the possibility that GH insufficiency could be the missing link between all these factors and COVID-19 severity [23]. Several reasons support this assumption: GH is physiologically involved in developing and maintaining the immune system [40] and different lymphocytes express GH receptors [47].

Furthermore, liver steatosis appears to contribute to the worse prognosis of COVID-19 patients [11].

It is tempting to speculate that the clinical severity of SARS-CoV-2 infections may be related to the interplay among ferritin metabolism, obesity, hepatic steatosis, and the GH/IGF-1 axis. To confirm our hypothesis, we conducted a retrospective analysis of the GH/IGF-1 axis in COVID-19 patients and evaluated several baseline factors to see which were associated with worse outcomes. We found that IGF-1 can be a simple and accurate tool to predict mortality or the need for ventilation in patients with COVID-19, as seen by other authors [48,49,50]. Our results are partially discordant with those of Sandeep Dhindsa et al. [10] but these authors did not consider the interference of sex and age on the IGF-1 values and did not normalize IGF-1 values for these two parameters nor expressed them as zSDSs. However, we are aware that in view of the small sample size and the complex clinical picture of SARS-CoV-2 infection, the single IGF-1 value is not sufficient to discriminate all patients hospitalized for COVID-19. Nevertheless, it can still provide a clue for the clinician to better classify the patient, especially when evaluated together with other parameters such as ferritin or steatosis indices. Understanding the link between obesity and severe COVID-19 outcomes requires further research and other studies concerning the discovery of new prognostic factors should be encouraged.

IGF-1 is a key predictive factor for metabolic alterations in obesity as it represents a mitogenic hormone involved in processes like growth, angiogenesis, and differentiation [51]. In individuals with obesity, lower IGF-1 serum levels and a blunted response to GH-stimulating dynamic tests are associated with more significant metabolic impairment and even morpho-functional cardiological alterations [52]. Higher serum IGF-1 in obese patients correlates with lower inflammatory patterns and better skeletal health [53]. Moreover, in a preliminary analysis with a machine learning approach, our group found that IGF-1 plays a crucial role in the pathogenesis of the metabolic derangement observed in many patients with obesity [54].

Furthermore, our work confirms that ferritin levels and liver damage (in the form of hepatic steatosis) worsens the prognosis of COVID-19, as already shown by Bucci and colleagues [55] and by our group [3,4]. A further step was to develop a model that could predict the risk of lung impairment and the need for oxygen therapy in COVID-19 patients by integrating clinical characteristics and laboratory parameters for each patient.

Intriguingly, with a multivariate regression analysis, we demonstrated that ferritin, hepatic attenuation, and zSDS-IGF-1 are all independent predictors of COVID-19 severity. The findings herein are consistent with data from other clinical trials [4,5,56]. A possible mechanism linking somatotropic axis defect and COVID-19 prognosis may be related to enhanced FXR signaling, with alterations of ACE2 expression and iron metabolism [19,20].

COVID-19 places a considerable burden on healthcare economics and thus requires significant adaptation of hospital facilities. In light of this, the results of our study indicate that, in addition to clinical assessment, the use of the rapid ferritin test and the IGF-1 assay can be a helpful tool to reliably determine whether admission to an intensive care unit is necessary for a specific patient. Faster identification of the most critical patients could save time in their clinical management and avoid overcrowding in the emergency department. Therefore, these results could have a positive impact on both the patient and the national economy.

Our study has several limitations. First, this is a single-center study, with a relatively small cohort of patients included; therefore, data should be confirmed in a larger trial. Second, the study’s retrospective design does not allow any cause-and-effect relationship to be established. Moreover, liver damage was assessed only at the time of admission, and it has not been possible to determine whether it persisted after the acute phase of COVID-19. Despite its limitations, the present study, performed in a population that underwent chest CT scan, blood gas analysis, and accurate biochemical evaluation, is the first one that investigates the somatotropic axis in relation to hepatic steatosis in SARS-CoV-2-infected patients and could provide a fascinating insight into the link between GH/IGF-1 axis impairment and lung disease severity in COVID-19 patients.

## 5. Conclusions

This retrospective study evaluated the GH/IGF-1 axis status at the time of hospital admission in a cohort of patients hospitalized for COVID-19. Our data show that the severe forms of COVID-19 are associated with an impairment of the GH–IGF-1 axis, along with higher BMI, higher ferritin levels, and reduced liver attenuation on non-contrast CT, all of which are associated with an increased likelihood of ventilation or death. In conclusion, our results may shed light on the possible role of IGF-1 as a new metabolic health parameter capable of effectively predicting the development of more severe forms of COVID-19. Thus, the finding of low serum IGF-1 levels and zSDS–IGF-1 in COVID-19 patients at admission should predict more severe disease and could lead to more rapid and necessary therapeutic measures, modulating FXR expression [56]. In addition, studying the immune responses underlying the different clinical presentations of COVID-19 in relation to the GH/IGF-1 axis could unveil new targets for more effective treatments [57]. Additional studies are warranted to further explore the potential therapeutic appropriateness of GH replacement therapy targeting the IGF-1 pathway in specific subgroups of COVID-19 patients with low IGF-1 levels who, as shown here, are at high risk of developing more severe disease.

## Figures and Tables

**Figure 1 viruses-15-00488-f001:**
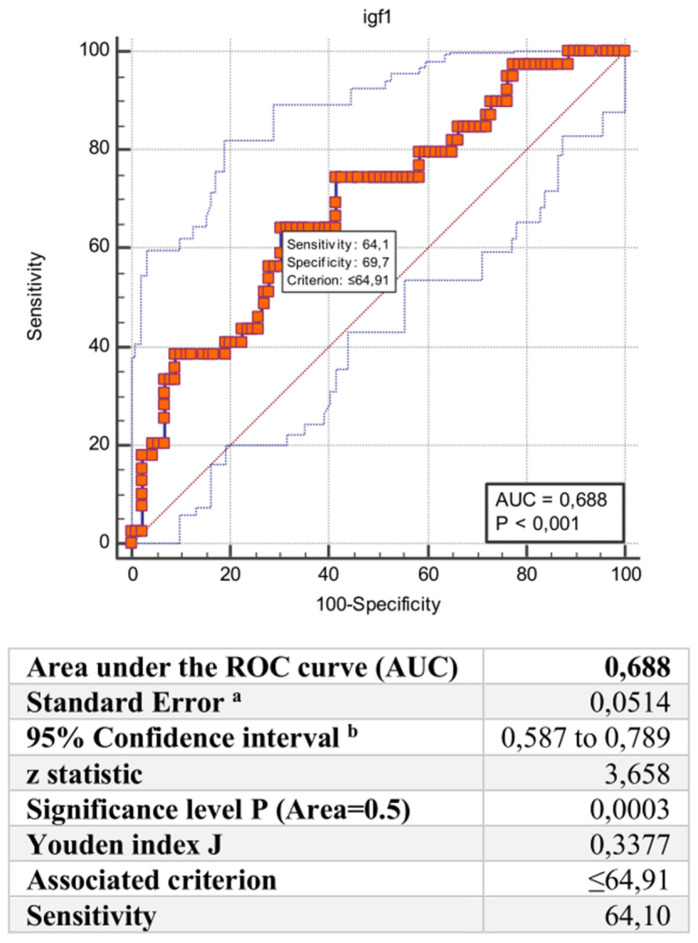
ROC analysis of serum IGF-1 levels (ng/dL) for the occurrence of ARDS/death among COVID-19 patients. ^a^ DeLong et al., 1988 [43]; ^b^ AUC ± 1.96 SE.

**Table 1 viruses-15-00488-t001:** Summary characteristics of the cohort at the time of hospital admission.

	143 Patients (68 Females–75 Males)			
Variables	Mean	S.D.	Minimum	Maximum
Age (years)	60.63	17.05	20.00	92.00
Liver attenuation (HU)	50.72	9.46	25.00	72.00
BMI (Kg/m^2^)	27.00	3.64	20.06	38.30
Ferritin (µg/L)	847.13	799.15	12.00	4000.00
CRP (mg/dL)	7.16	9.14	0.09	45.40
DD (µg/dL)	1242.37	1216.64	169.00	4610.00
Fibrinogen (mg/dL)	498.65	87.25	241.00	829.00
Platelet × 10^9^/L	214.12	89.56	40.00	656.00
LDH (UI/L)	324.41	137.51	132.00	874.00
Leukocytes × 10^3/^µL	7.09	3.74	2.00	22.23
Neutrophils × 10^3/^µL	5.44	3.47	1.12	19.38
Lymphocytes × 10^3/^µL	1.08	0.69	0.19	5.44
Monocytes × 10^3/^µL	0.43	0.59	0.05	6.10
Glycemia (mg/dL)	123.79	49.53	72.00	397.00
HbA1c (%)	5.76	0.98	4.50	9.00
Creatinine (mg/dL)	0.97	0.53	0.40	5.00
Na (mM/L)	137.47	5.17	117.00	154.00
K (mM/L)	4.43	3.75	2.91	38.00
Ca (mg/dL)	8.70	0.74	7.40	10.90
AST (mU/mL)	36.81	29.46	3.86	259.00
ALT (mU/mL)	43.49	122.57	7.00	1379.00
GGT (U/L)	46.48	70.37	0.38	529.00
CPK (U/L)	154.03	174.11	20.00	779.00
GH (ng/mL)	0.92	1.06	0.05	6.52
IGF-1 (ng/dL)	97.15	63.40	20.90	426.50
zSDS–IGF-1	–2.62	1.64	–5.30	2.37
P/F ratio	310.46	115.14	58.00	590.00
HSI	36.48	6.87	27.15	69.99
FIB–4	2.31	1.86	0.06	12.62
Number of comorbidities	1.01	1.24	0.00	7.00

**Abbreviations:** HU, Hounsfield unit; BMI, body mass index; CRP, C-Reactive Protein; DD, D–dimer; LDH, lactate dehydrogenase; HbA1c, Glycated hemoglobin; Na, sodium; K, potassium; Ca, calcium; AST, aspartate aminotransferase; ALT, alanine aminotransferase; γ GT, Gamma-glutamyl transferase; CPK, Creatine phosphokinase; GH, growth hormone; IGF-1, insulin-like growth factor 1; zSDS–IGF-1, insulin-like growth factor 1 standard deviation score; P/F ratio, arterial oxygen partial pressure to fractional inspired oxygen; HIS, hepatic steatosis index; FIB-4, Fibrosis-4 score.

**Table 2 viruses-15-00488-t002:** One-way analysis of variance of laboratory, clinical, and CT measurements according to the severity of disease (pulmonary impairment in CT and respiratory failure)—0 = mild: no CT alterations; 1 = moderate: changes in CT scan, no oxygen; 2 = severe: CT scan alterations plus oxygen; 3 = critical: CT abnormalities plus intensive care unit (ICU) admission for ARDS.

	Group 0	Group 1	Group 2	Group 3		
Variables	Mean (SD)	Mean (SD)	Mean (SD)	Mean (SD)	F	P
Number (F/M)	17 (10/7)	26 (9/17)	37 (22/15)	63 (27/36)	1.63	0.19
Age (years)	49.82 (18.75)	61.80 (15.23)	60.84 (17.78)	62.83 (16.21)	**2.75**	**0.05**
Liver attenuation (HU)	58.69 (6.02)	51.61 (10.43)	51.42 (6.24)	46.86 (10.03)	**8.04**	**0.00**
BMI (Kg/m^2^)	24.37 (3.52)	25.67 (3.30)	26.77 (2.49)	28.47 (3.78)	**5.27**	**0.00**
Ferritin (µg/L)	234.12 (184.92)	478.32 (383.11)	733.99 (677.99)	1203.92 (897.70	**11.91**	**0.00**
CRP (mg/dL)	1.57 (3.49)	3.89 (3.60)	7.74 (9.56)	9.59 (10.50)	**4.95**	**0.00**
DD (µg/dL)	468.29 (269.95)	1186.84 (1148.91)	1392.64 (1208.16)	1399.61 (1358.70)	**2.91**	**0.04**
Fibrinogen (mg/dL)	429.12 (82.49)	475.84 (77.52)	493.17 (89.19)	531.29 (77.53)	**7.96**	**0.00**
Platelet × 10^3/^µL	207.88 (62.60)	193.96 (70.70)	221.54 (76.67)	220.85 (109.32)	0.64	0.59
LDH (UI/L)	211.31 (69.71)	292.41 (109.41)	308.34 (118.60)	376.96 (145.58)	**8.08**	**0.00**
Leukocytes × 10^9^/L	6.30 (3.25)	6.86 (3.44)	6.75 (2.94)	7.70 (4.33)	0.93	0.43
Neutrophils × 10^9^/L	4.51 (3.20)	4.92 (2.78)	5.01 (2.69)	6.23 (4.07)	1.86	0.14
Lymphocytes × 10^9^/L	1.23 (0.68)	1.13 (0.61)	1.20 (0.93)	0.95 (0.52)	1.40	0.25
Monocytes × 10^9^/L	0.35 (0.15)	0.55 (0.68)	0.53 (0.96)	0.36 (0.19)	1.15	0.33
Glycemia (mg/dL)	132.50 (35.77)	105.88 (18.26)	117.53 (33.06)	130.66 (64.82)	1.38	0.25
Glycated Hb %	5.3 (0.27)	5.53 (0.32)	5.71 (0.44)	5.85 (1.25)	0.20	0.89
Creatinine (mg/dL)	0.89 (0.56)	1.14 (0.98)	0.85 (0.329	0.99 (0.41)	1.36	0.26
Na (mM/L)	137.07 (2.99)	138.71 (5.94)	138.59 (3.36)	136.68 (6.15)	1.16	0.33
K (mM/L)	5.33 (5.20)	3.91 (0.57)	5.12 (6.34)	3.94 (0.55)	1.00	0.40
Ca (mg/dL)	8.60 (1.19)	8.00 (0.42)	8.93 (0.44)	8.68 (0.77)	0.91	0.45
AST (mU/mL)	26.94 (9.39)	29.55 (10.43)	31.26 (19.69)	44.76 (38.96)	**3.01**	**0.03**
ALT (mU/mL)	26.76 (23.62)	29.65 (16.31)	31.33 (18.32)	60.17 (181.05)	0.65	0.59
GGT (U/L)	20.40 (9.23)	38.44 (17.31)	61.74 (121.49)	47.90 (46.13)	0.78	0.51
CPK (U/L)	68.33 (20.80)	110.38 (60.15)	134.96 (135.87)	180.62 (203.27)	1.42	0.24
GH (ng/mL)	2.27 (5.46)	1.39 (1.53)	0.99 (1.23)	0.68 (0.68)	2.36	0.07
IGF-1 (ng/dL)	141.51 (61.62)	123.74 (89.17)	92.97 (49.44)	76.68 (48.52)	**6.85**	**0.00**
zSDS–IGF-1	–0.51 (2.04)	–0.48 (3.19)	–2.01 (1.51)	–2.43 (1.47)	**5.89**	**0.00**
P/F ratio	420.56 (54.29)	371.11 (71.57)	329.97 (95.59)	239.85 (109.83)	**18.97**	**0.00**
HSI	33.79 (2.93)	33.80 (3.12)	37.56 (5.27)	37.55 (8.66)	1.45	0.24
FIB–4	1.40 (0.58)	2.09 (1.10)	1.88 (1.34)	2.35 (1.46)	**2.63**	**0.05**
Number of comorbidities	0.50 (0.51)	0.61 (0.87)	1.03 (1.03)	1.30 (1.60)	1.94	0.13

**Abbreviations:** HU, Hounsfield unit; BMI, body mass index; CRP, C-Reactive Protein; DD, D–dimer; LDH, lactate dehydrogenase; HbA1c, Glycated hemoglobin; Na, sodium; K, potassium; Ca, calcium; AST, aspartate aminotransferase; ALT, alanine aminotransferase; γ GT, Gamma-glutamyl transferase; CPK, Creatine phosphokinase; GH, growth hormone; IGF-1, insulin-like growth factor 1; zSDS–IGF-1, insulin-like growth factor 1 standard deviation score; P/F ratio, arterial oxygen partial pressure to fractional inspired oxygen; HIS, hepatic steatosis index; FIB-4, Fibrosis-4 score.

**Table 3 viruses-15-00488-t003:** Student’s *t*-test analysis of laboratory, clinical, and CT measurements according to the occurrence = 0 or non-occurrence = 1 of ARDS/death.

	Mean	S.D.	Mean	S.D.	*p*
	0	0	1	1	
Number (F/M)	83 (43/40)		59 (25/34)		
Age (years)	58,140	17,395	63,980	16,188	0.044
Liver attenuation (HU)	51,830	10,080	48,160	8470	0.030
BMI (Kg/m^2^)	26,010	3198	28,710	3754	**0.000**
Ferritin (µg/L)	559,330	554,794	1229,120	911,059	**0.000**
CRP (mg/dL)	5150	7283	10,000	10,720	**0.001**
DD (µg/dL)	1096,410	1094,369	1451,490	1368,123	0.097
Fibrinogen (mg/dL)	481,000	83,847	524,410	87,028	**0.004**
Platelet × 10^9^/L	209,710	73,770	221,890	109,861	0.436
LDH (UI/L)	290,150	116,370	372,810	148,903	**0.000**
Leucocytes × 10^9^/L	6780	3260	7630	4301	**0.183**
Neutrophils × 10^9^/L	4950	2875	6210	4100	**0.034**
Lymphocytes × 10^9^/L	1190	0763	0920	0521	**0.018**
Monocytes × 10^9^/L	0490	0741	0350	0194	0.149
Glycemia (mg/dL)	116,420	31,295	134,730	66,943	**0.049**
Glycated Hemoglobin (%)	5480	510	6080	1270	0.068
Creatinine (mg/dL)	0930	0604	1020	0419	0.357
Na (mM/L)	138,250	3331	136,480	6820	0.073
K (mM/L)	4780	4970	3970	0560	0.262
AST (mU/mL)	30,600	15,948	44,940	40,339	**0.005**
ALT (mU/mL)	29,570	18,736	63,280	189,207	0.127
Ca (mg/dL)	8760	0696	8610	0812	0.582
GGT (U/L)	45,230	83,849	48,430	46,435	0.855
CPK (U/L)	112,860	104,999	190,860	211,875	**0.036**
GH (ng/mL)	1400	2774	0,640	0709	0.060
IGF-1 (ng/dL)	109,680	69,411	79,760	49,500	**0.008**
zSDS–IGF-1	−1380	2243	−2330	1522	**0.028**
P/F ratio	352,680	98,920	243,950	108,527	**0.000**
HSI	35,090	4458	38,570	9104	**0.037**
FIB–4	1980	1431	2860	2383	**0.039**
Number of comorbidities	0760	0893	1480	1671	**0.007**

**Abbreviations:** HU, Hounsfield unit; BMI, body mass index; CRP, C-Reactive Protein; DD, D–dimer; LDH, lactate dehydrogenase; HbA1c, Glycated hemoglobin; Na, sodium; K, potassium; Ca, calcium; AST, aspartate aminotransferase; ALT, alanine aminotransferase; γ GT, Gamma-glutamyl transferase; CPK, Creatine phosphokinase; GH, growth hormone; IGF-1, insulin-like growth factor 1; zSDS–IGF-1, insulin-like growth factor 1 standard deviation score; P/F ratio, arterial oxygen partial pressure to fractional inspired oxygen; HIS, hepatic steatosis index; FIB-4, Fibrosis-4 score.

**Table 4 viruses-15-00488-t004:** Multivariate regression analysis to evaluate independent predictors of COVID-19 severity (Groups 0–3).

	R = 0.56452776 R^2^ = 0.31869159 Adjusted R^2^ = 0.29179784 F (3.76) = 11,850 p					
N = 143	b*	S.E. of b*	b	S.E. of b	t (76)	*p*-Value
Intercept			3.25839	0.61712	5.27995	**0.00000**
Ferritin, µg/L	0.36698	0.09620	0.00029	0.00008	3.81465	**0.00028**
Liver attenuation, HU	−0.363571	0.10228	−0.038050	0.01071	−3.55451	**0.00066**
zSDS–IGF-1	−0.214087	0.10080	−0.143060	0.06736	−2.12378	**0.03694**

**Abbreviations:** HU, Hounsfield unit; zSDS–IGF-1, insulin-like growth factor 1 standard deviation score.

## Data Availability

Data will be made available upon reasonable request to the corresponding author.

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
