# Peer review of "Obesity-Associated Hepatic Steatosis, Somatotropic Axis Impairment, and Ferritin Levels Are Strong Predictors of COVID-19 Severity"

_viruses, 2023, doi:10.3390/v15020488_

Round 1
Reviewer 1 Report
This retrospective analysis of a covid-19 cohort tries to explore a hypothesis concerning its pathophysiology. Although a retrospective analysis can never provide the definite answer, these authors have presented interesting data that can add to our knowledge on this subject. The paper is well written, is clear in its results and addresses the pro's and con's in the discussion. I have no further questions.
Author Response
Dear reviewer thank you for your kind comments and appreciation of our article.
Reviewer 2 Report
Comments to Authors
This study showed that an impairment of the GH–IGF1 axis in the severe forms of COVID–19 along with higher BMI, higher ferritin levels and reduced liver attenuation on non–contrast CT, all of which are associated with an increased likelihood of ventilation or death.
Authors are kindly requested to emphasize the current concepts about these issues in the context of recent knowledge and the available literature. This articles should be quoted in the References list.
References
1. Diet-Induced Obesity and NASH Impair Disease Recovery in SARS-CoV-2-Infected Golden Hamsters. Viruses. 2022; 14 (9): 2067. Published 2022 Sep 17. doi:10.3390/v14092067.
2. Association of COVID-19 with hepatic metabolic dysfunction. World J Virol. 2022; 11 (5): 237-251. doi:10.5501/wjv.v11.i5.237.
Author Response
Dear reviewer thank you for your comments. As pointed out, we have added the two relevant missing references in the introduction which helped us to provide an adequate background for our study. Please see referencese [27] and [28].
Reviewer 3 Report
In this article the authors searched for markers that could explain the complex clinical variability of COVID-19. They carried out a retrospective study, which included hospitalized patients infected with SARS-CoV-2, and divided the patients in four groups depending on severity of pulmonary impairment on the CT and respiratory failure. The authors concluded that ferritin, IGF-1 and liver steatosis were significant predictors for severity and poor prognosis in obese patients.
This article is in general well written and, in my opinion, the retrospective study design is correct, although sample size (n=143, with 17, 26, 37, and 63 for groups 0, 1, 2, and 3 respectively) might be limited, mostly for groups 0 and 1; however I understand that hospitalized patients would be mostly in groups 2 and 3, and the authors acknowledge the need to confirm the data with a larger trial.
Line 50 (Introduction section) indicate what NASH stands for (Non-Alcoholic Steato-Hepatitis?)
Line 129 (section 2.3) and line 254 (Discussion section), probably the authors meant “sex” instead of “gender”? According to the WHO, sex refers to “the different biological and physiological characteristics of males and females, such as reproductive organs, chromosomes, hormones, etc.”, whereas gender refers to "the socially constructed features of women and men – such as norms, roles and relationships of and between groups of women and men. It varies from society to society and can be changed..." please revise the concept if necessary.
Please discuss further the predictive usefulness (64.1% sensitivity and 69.7% specificity) of the cut-off of 64.91 ng/dl of serum IGF-1 in the context of a complex disease as COVID-19 with a variety of symptoms and severity degrees. Are these values (considering the sample size) good enough to consider IGF-1 a predictive marker of severity? Is this marker valid only for obese patients or it may work for other non-obese patients?
Line 201 says “Figure 2” but I believe is “Figure 1”.
Table 3: indicate if 0=non-occurrence, and 1=occurrence
The results indicate that ferritin, hepatic attenuation, and zSDS-IGF1 are independent predictors of COVID-19 severity in obese patients. Which would be the specific recommendation(s) to the clinic regarding the use and predictive value of these markers? In other words, which are the potential benefits to the clinic?
Even if the patients were not directly involved in any phase of the study, and a waiver of informed consent was granted, please indicate the number of ethical approval provided by the Institutional Review Board.
Author Response
Dear reviewer many thanks for your positive comments concerning our manuscript which were intellectually stimulating and helped us to improve it considerably. We carefully addressed your comments and revised our manuscript accordingly. Changes were highlighted using the track changes mode in the revised manuscript.Please find a point-by-point response below including all the corrections.
- Line 50 (Introduction section) indicate what NASH stands for (Non-Alcoholic Steato-Hepatitis?) Thank you, we have made this change.
- Line 129 (section 2.3) and line 254 (Discussion section), probably the authors meant “sex” instead of “gender”? According to the WHO, sex refers to “the different biological and physiological characteristics of males and females, such as reproductive organs, chromosomes, hormones, etc.”, whereas gender refers to "the socially constructed features of women and men – such as norms, roles and relationships of and between groups of women and men. It varies from society to society and can be changed..." please revise the concept if necessary.
We apologise for using the wrong word. We have made the appropriate changes.
Please discuss further the predictive usefulness (64.1% sensitivity and 69.7% specificity) of the cut-off of 64.91 ng/dl of serum IGF-1 in the context of a complex disease as COVID-19 with a variety of symptoms and severity degrees. Are these values (considering the sample size) good enough to consider IGF-1 a predictive marker of severity? Is this marker valid only for obese patients or it may work for other non-obese patients?
Thank you for the constructive comment. We agree that the limited number of patients included in our study may reduce the prognostic value of the IGF-1 cut-off in our analysis, and we have listed this among the limits of our study. However, the ROC curve confirms that IGF-1 may be a relevant parameter in the early identification of patients who are at higher risk of developing severe forms of COVID-19.
To the best of our knowledge, there are no specific data in the literature on IGF-1 levels in the non-obese population with COVID-19. The prognostic value of IGF-1 while remaining valid in the normal-weight population is more accurate in the population with overweight or obesity, which constitutes the category of patients with more severe forms of the disease. We explored this aspect further in the discussion.
- Line 201 says “Figure 2” but I believe is “Figure 1”.
Thank you, we have made this change.
- Table 3: indicate if 0=non-occurrence, and 1=occurrence
Thank you, we have specified this.
- The results indicate that ferritin, hepatic attenuation, and zSDS-IGF1 are independent predictors of COVID-19 severity in obese patients. Which would be the specific recommendation(s) to the clinic regarding the use and predictive value of these markers? In other words, which are the potential benefits to the clinic?
As suggested, we further discussed possible clinical implications of the results of our study. The new parameters identified could help in the early identification of patients who are at greater risk of developing severe forms of COVID-19. Therefore, these results could have a positive impact on both the patient and the national economy.
- Even if the patients were not directly involved in any phase of the study, and a waiver of informed consent was granted, please indicate the number of ethical approval provided by the Institutional Review Board.
The study was approved by the local IRB (prot. 5475/19) and conducted in accordance with the Declaration of Helsinki and Good Clinical Practice. Written informed consent was obtained from all study participants before enrollment.
Round 2
Reviewer 2 Report
Several improvements and clarifications have been made to the study.Reviewer 3 Report
In this revised version of the manuscript the authors answered all my questions and I appreciate the effort. I have no further comments.